# Evaluation of the genetic risk for COVID-19 outcomes in COPD and differences among worldwide populations

Rui Marçalo[1,2]*, Sonya Neto[1], Miguel Pinheiro[1], Ana J. Rodrigues[3], Nuno Sousa[3], Manuel A. S. Santos[1], Paula Simão[4], Carla Valente[5], Lília Andrade[5], Alda Marques[2], Gabriela R. Moura[1]

1 Department of Medical Sciences, Genome Medicine Laboratory, Institute of Biomedicine—iBiMED, University of Aveiro, Aveiro, Portugal, 2 Lab3R-Respiratory Research and Rehabilitation, School for Health Sciences (ESSUA) and Institute of Biomedicine (iBiMED), University of Aveiro, Aveiro, Portugal, 3 Life and Health Sciences Research Institute (ICVS), School of Medicine, University of Minho–Braga, Portugal, 4 Pulmonology Department, Unidade Local de Saúde de Matosinhos—Porto, Porto, Portugal, 5 Pulmonology Department, Centro Hospitalar do Baixo Vouga–Aveiro, Aveiro, Portugal

* ruifilipemarcalo@ua.pt

**Data Availability Statement:** All relevant data are within the paper and its Supporting Information files.

**Funding:** This work was supported by FEDER (Fundo Europeu de Desenvolvimento Regional)

## Abstract

### Background

Populations seem to respond differently to the global pandemic of severe acute respiratory syndrome coronavirus 2. Recent studies show individual variability in both susceptibility and clinical response to COVID-19 infection. People with chronic obstructive pulmonary disease (COPD) constitute one of COVID-19 risk groups, being already associated with a poor prognosis upon infection. This study aims contributing to unveil the underlying reasons for such prognosis in people with COPD and the variability in the response observed across worldwide populations, by looking at the genetic background as a possible answer to COVID-19 infection response heterogeneity.

### Methods

SNPs already associated with susceptibility to COVID-19 infection (rs286914 and rs12329760) and severe COVID-19 with respiratory failure (rs657152 and rs11385942) were assessed and their allelic frequencies used to calculate the probability of having multiple risk alleles. This was performed on a Portuguese case-control COPD cohort, previously clinically characterized and genotyped from saliva samples, and also on worldwide populations (European, Spanish, Italian, African, American and Asian), using publicly available frequencies data. A polygenic risk analysis was also conducted on the Portuguese COPD cohort for the two mentioned phenotypes, and also for hospitalization and survival to COVID-19 infection.

### Findings

No differences in genetic risk for COVID-19 susceptibility, hospitalization, severity or survival were found between people with COPD and the control group (all p-values > 0.01),

funds through the COMPETE 2020, Operational Programme for Competitiveness and Internationalization (POCI) (POCI-01-0145-FEDER-028806; POCI-01-0145-FEDER-016428), CENTRO 2020 CENTRO 2020 (CENTRO-01-0246-FEDER-000018) and by Portuguese national funds via Fundação para a Ciência e a Tecnologia, I.P. (FCT) under the projects PTDC/DTP-PIC/2284/2014; PTDC/SAU-SER/28806/2017; PTDC/BIA-MIC/31849/2017, and the PhD fellowship UI/BD/151337/2021. The iBiMED is supported by FCT funds under UIDP/04501/2020.

**Competing interests:** The authors have declared that no competing interests exist.

either considering risk alleles individually, allelic combinations or polygenic risk scores. All populations, even those with European ancestry (Portuguese, Spanish and Italian), showed significant differences from the European population in genetic risk for both COVID-19 susceptibility and severity (all p-values < 0.0001).

## Conclusion

Our results indicate a low genetic contribution for COVID-19 infection predisposition or worse outcomes observed in people with COPD. Also, our study unveiled a high genetic heterogeneity across major world populations for the same alleles, even within European sub-populations, demonstrating the need to build a higher resolution European genetic map, so that differences in the distribution of relevant alleles can be easily accessed and used to better manage diseases, ultimately, safeguarding populations with higher genetic predisposition to such diseases.

## Introduction

There is increasing evidence about both individual and populational variability in the susceptibility and disease behaviour of patients with severe acute respiratory syndrome coronavirus 2 (SARS-CoV-2) infection, ranging from asymptomatic to severe respiratory failure and need for prolonged mechanical ventilation [1]. The underlying causes of this spectrum of clinical outcomes remain elusive. Determining the genetic and environmental risk factors capable of originating such discrepancies is important to identify those at most risk, enabling to protect them from infection, reduce hospitalizations and decrease mortality.

Recently, the human genetics community came together and launched the COVID-19 Host Genetics Initiative [2], an unprecedent worldwide collaboration whose mission is to generate, share and identify genetic variants associated with COVID-19 susceptibility, severity and other clinically relevant outcomes. It is hoped that this international effort may help identify groups of people at high or low risk, as well as to generate hypotheses for therapies, and contribute to an advance in the global knowledge of COVID-19 physiopathology.

Two large-patient cohorts from Italy and Spain, epicentres of the COVID-19 first wave of pandemic in Europe, have been analysed in a genome-wide association study (GWAS) that looked for genetic variants associated with the occurrence of severe COVID-19 with respiratory failure [3]. This study identified two genetic variants significantly associated to this phenotype, namely: rs11385942 at locus 3p21.31 and rs657152 at locus 9q34.2. Additionally, three independent studies, aimed at finding if patients' genetic background could be responsible for the differences in susceptibility to COVID-19 infection [4–6], identified 3 genetic variants associated with COVID-19 positive testing: rs286914 at locus 11p13, rs12329760 at locus 21q22.3 and rs41303171 at locus Xp22.2.

Chronic obstructive pulmonary disease (COPD) is the third leading cause of mortality worldwide [7]. The current evidence suggests that COVID-19 incidence is not higher in people with COPD, however, chronic inflammation of the small airways, higher rates of hospitalization and admission to intensive care unit (ICU), as well as invasive mechanical ventilation [8], show that the COPD sub-population may be particularly vulnerable to severe COVID-19 [9,10]. In order to determine what is causing people with COPD to respond more severely, both genetical and environmental factors need to be tackled, further developing our

knowledge on the pathophysiology of COPD itself and the reasons behind a worse evolution of COVID-19 on people with COPD [8]. Therefore, we aimed to explore 1) the genetic risk of people with COPD for increased COVID-19 incidence and severity and 2) how the same risk alleles are distributed across the European population, its sub-populations (Portuguese, Spanish and Italian); and the other major worldwide populations (African, American and Asian), estimating the global distribution of people with an increased genetic risk for COVID-19 infection and severe COVID-19.

## Materials and methods

### Recruitment of participants

In this study, two cohorts were used. The Baixo Vouga cohort [i.e. healthy controls and individuals with COPD] was used for the case-control study, whereas the Minho cohort [i.e. community-dwelling individuals] was used for validation.

The Baixo Vouga cohort recruitment was performed by physicians, from hospitals and primary healthcare centres of the Centre Region of Portugal. A set of inclusion criteria was established for people with COPD, i.e., having a diagnosis of COPD according to Global Initiative for Chronic Obstructive Lung Disease GOLD criteria [11]; being in a stable state (i.e., no acute exacerbations in the previous month), and able of giving informed consent. Control group individuals (sex- and age- matched to people with COPD) were recruited from routine appointments in primary health care centres and senior universities. They were considered eligible if they did not have a diagnosed respiratory disease and were able to give informed consent. Exclusion criteria for both groups were set for the presence of severe cardiac, musculoskeletal, or neuromuscular diseases, cognitive impairment or a history of neoplasia or immune disease that would interfere with patients' collaboration in data collection or interpretation. Therefore, people with the most prevalent age-related conditions, e.g., controlled hypertension, hypercholesterolaemia were included in the study since according to the World Health Organization definition, Health is a state of complete physical, mental and social wellbeing and not merely the absence of disease or infirmity [12]. Sociodemographic (age, sex), anthropometric (weight and height to compute body mass index), clinical data and saliva samples were collected with a structured protocol. Information on lung function was retrieved from clinical records or collected via spirometry.

The Minho cohort is part of a larger cohort randomly selected from the north of Portugal [13,14]. Primary exclusion criteria included participants diagnosed with dementia, who had a stroke, renal failure or overt thyroid pathology. The cohorts were established according to the ethical principles of the Declaration of Helsinki and experiments were approved by the Portuguese ethical committee (Comissão Nacional de Proteção de Dados) and local ethic review boards (Administração Regional de Saúde do Centro; Centro Hospitalar do Baixo Vouga; Unidade Local de Saúde de Matosinhos; Centro Hospitalar do Médio Ave; Hospital Distrital da Figueira da Foz; Hospital Escola Braga, Braga; Centro Hospitalar do Alto Ave, Guimarães; and Unidade Local de Saúde do Alto Minho, Viana do Castelo/Ponte de Lima). All participants provided voluntary and informed written consent.

### Individual genotyping

DNA for genotyping was extracted from saliva samples (Baixo Vouga cohort) or peripheral blood mononuclear cells (Minho cohort) using QIAamp DNA Mini Kit (QIAGEN) and following the kit's protocol with slight modifications. Samples from the Baixo Vouga cohort were genotyped using the Infinitum Global Screening Array-24 v1.0 while the Minho cohort was genotyped using the Neuro Consortium Array (Infinium Core-24+ v1.2, Illumina), following

the Illumina Infinitum HTS Assay protocols. Quality control (QC), of both cohorts, was performed using PLINK 1.9 [15] software following standard protocols [16,17]. Briefly, samples having a call rate lower than 95%; gender discrepancies; divergent ancestry; heterozygosity rate higher than three standard deviations away from the mean rate; and third-degree related individuals were excluded. Additionally, SNPs with missing rate higher than 5%; showing deviations from Hardy-Weinberg equilibrium; with a minor allele frequency lower than 5%; and displaying distinct missing rates when comparing cases with controls were also excluded [16]. Subsequently genotypes were imputed using the Michigan Imputation Server [18] and submitted to the same QC filters [19–21]. A total of 498 samples and 6.838.946 SNPs in the Baixo Vouga cohort and 380 samples and 5.389.594 SNPs in the Minho cohort passed QC and were used in this work.

## Genetic risk assessment

SNPs significantly associated with susceptibility to COVID-19 infection [4–6] (rs286914 and rs12329760) and severe COVID-19 with respiratory failure [3] (rs657152 and rs11385942) were selected from the literature (S1 Table). Allelic frequencies were used to calculate the probability of having multiple risk alleles in each population (European, Portuguese, Spanish, Italian, African, American and Asian) (S2 Table). For each SNP, genotypes where coded as 0,1,2 depending on the number of risk alleles, with the genetic risk category representing the total count of alleles for each pair of SNPs (score range 0–4). Data for the Portuguese(n = 623), Spanish (n = 9761) and Italian (6363) populations correspond to observed values extrapolated to 1 million, whereas data for remaining populations correspond to estimations (also to 1 million) based on the published effect allele frequencies (rs286914, rs12329760 and rs11385942 data were obtained from gnomAD-Genome project [22], while rs657152 data was obtained from the ALFA project] [23]. Prior to estimations, Hardy-Weinberg equilibrium was assessed and validated (susceptibility SNPs: p-value = 0.80; severity SNPs: p-value = 0.29).

Polygenic risk scores (PRS) offer a quantifiable metric of an individual's inherited risk based on the cumulative impact of several common polymorphisms and can be used to test the genetic propensity of an individual to a wide range of diseases. PRS for severe COVID-19 with respiratory failure [3], susceptibility to COVID-19 infection [24] and COVID-19 related hospitalization [24] and survivability [24] were calculated using publicly available summary statistics. All summary statistics used resulted from studies conducted among participants of European ancestry. PRSice software [25] was used for the calculation of individual PRS within our cohorts, following a standard protocol [26,27]. Analyses were adjusted for the same covariables as the original GWAS [2,3] (gender, age, and the first 10 principal components).

## Statistical analysis

Statistical analyses were performed using IBM SPSS Statistics, version 25 (IBM Corporation, Armonk, NY, USA) and plots created using GraphPad Prism, version 6 (GraphPad, San Diego, CA, USA). Differences between groups' effect allele frequencies (both COPD vs control and European vs remaining world populations) were assessed by means of a Chi-Square test. Chi-Square test was also used to assess differences in proportions between populations (European vs every other population) for the cumulative risk of having two SNPs, for either COVID-19 infection susceptibility or severe COVID-19 with respiratory failure. Given the magnitude of the differences observed in the analyses of European vs the remaining populations, an adjusted residues analysis [28] was performed to specify which population and/or

risk category (0 to 4) was being the most impactful on the statistical significance.

$$Adjusted\ residue = \frac{(observed - expected)}{\sqrt{expected * \left(1 - \frac{Row\ Marginal}{n}\right) * \left(1 - \frac{Column\ Marginal}{n}\right)}}$$

The normality of polygenic risk scores data was assessed using the Shapiro–Wilk test. Polygenic risk scores of control and COPD groups were compared by means of an unpaired student's t test (susceptibility, hospitalization and survivability), whereas for severity, a Mann-Whitney test was used. Statistical significance level for every analysis was set at 0.01.

## Results

### Cohort characterization

The Baixo Vouga cohort consisted of 498 people (243 controls and 255 individuals with COPD). Recruitment was performed so that groups were sex- and age- matched (male: 72.4% vs 79.6%, p-value = 0.06; age: 67 [60, 72] vs 68 [61, 74] years old, p-value = 0.044; control and COPD, respectively). All other measures were significantly different between both groups (body mass index: 27.32 [24.91, 29.75] vs 25.97 [23.44, 29.73] Kg/m$^2$, p-value < 0.01; FEV$_1$: 2.58 [2.11, 3.06] vs 1.32 [0.94, 1.81] litters, p-value < 0.01; FEV$_1$/FVC: 83.90 [78.01, 89.02] vs 53.02 [41.24, 61.94], p-value < 0.01; control and COPD, respectively). Detail characterization of the samples is shown in Table 1.

The Minho cohort was composed of 380 community-dwelling individuals, with a median age of 66 [53, 72] years, 48.2% (n = 183) male individuals and a median body mass index of 27.65 [25.1, 30.45] Kg/m$^2$ (all other metrics of Table 1 were not available for this cohort). Details characterization of the Minho cohort is shown in (S3 Table).

A new pseudo-cohort was created, composed of 623 individuals, and from now on designated as Portuguese cohort by merging both cohorts (control individuals from the Baixo Vouga cohort and all individuals from the Minho cohort) since no significant differences between original groups were found (S1 Fig). Comparison for SNP rs11385942 was not possible, as it was not genotyped or imputed in the Minho cohort. People with COPD, from the Baixo Vouga cohort, were not included given the disease-specific nature of the group. The Portuguese cohort median age was 66 [58, 72] years old, with 57.62% (n = 359) male individuals and a median BMI of 27.53 [25.08, 30.22] Kg/m$^2$. Detail characterization is shown in Table 1.

**Table 1. Sociodemographic, anthropometric and clinical characteristics of participants.**

| Characteristics | Baixo Vouga cohort | | Portuguese cohort |
|---|---|---|---|
| | COPD (n = 255) | Control (n = 243) | Control + Minho (n = 623) |
| Age (years) | 68 [61, 74] | 67 [60, 72] | 66 [58, 72] |
| Gender (Male), n (%) | 203 (79.61%) | 176 (72.43%) | 359 (57.62%) |
| Body mass Index (Kg/m$^2$) | 25.97 [23.44, 29.73] | 27.32 [24.91, 29.75] | 27.53 [25.08, 30.22] |
| FEV$_1$ (Litres) | 1.32 [0.94, 1.81] | 2.58 [2.11, 3.06] | NA |
| FEV$_1$/FVC | 53.02 [41.24, 61.94] | 83.90 [78.01, 89.02] | NA |

N (%)—number of individuals and corresponding percentage; remaining data is presented as medians with interquartile range in square brackets. FEV$_1$—Forced Expiratory Volume in 1-sec in litres; FVC—Forced Vital Capacity in litres; NA—no data available.

### Genetic risk assessment for COVID-19 associated phenotypes in people with COPD

**Single loci risk assessment.**   People with COPD (n = 255) did not display an increased frequency of risk alleles for COVID-19 infection susceptibility (rs286914 and rs12329760) or severe response to COVID-19 infection SNPs (rs657152 and rs11385942), when compared to the control group (n = 243) (p-value = 0.6; p-value = 0.55; p-value = 0.85; p-value = 0.72; respectively) (Fig 1). These results were validated on an external cohort from the Minho region of Portugal (n = 380), with no major differences emerging from the analyses (i.e. data from Minho was not different from the COPD nor the control group, data not shown).

**Bi-allelic risk assessment.**   The bi-allelic risk in our cohort was then calculated to determine the number of people that could be at a high risk for COVID-19 infection and a severe COVID-19 with respiratory failure, since each phenotype was previously associated with two SNPs. Homogeneity between the COPD and control groups was once again observed, with no differences in the number of people at low (0 risk alleles), moderate-low (1 risk allele), moderate (2 risk alleles), moderate-high (3 risk alleles) or high risk (4 risk alleles) for either the outcomes, (Fig 2). Most people were at low to moderate risk for COVID-19 infection, 244 for the COPD group (96%) and 233 for the control group (96%). Both groups had 10 people (4%) at a moderate-high risk. There was only 1 person, belonging to the COPD group, that had the highest risk for COVID-19 infection, being double-homozygous for the risk alleles (Fig 2A). Regarding the response to COVID-19 infection, 248 (97%) people with COPD and 239 (98%) people from the control group were at a low to moderate risk. There were 7 (3%) and 4 (2%) people, COPD and control groups respectively, with a moderate-high risk, but no one was a double-homozygous for the risk alleles, in either group (Fig 2B).

**Polygenic risk assessment.**   A complementary genetic analysis was conducted by calculating polygenic risk scores (PRSs) for four different phenotypes to gain further insight on the overall genetic risk for the onset of COVID-19 relevant phenotypes in people with COPD. Summary statistics from the COVID-19 Host Genetics Initiative were used to calculate PRS for: i) COVID-19 infection susceptibility, ii) COVID-19-related hospitalization, iii) severe COVID-19 with respiratory failure, and iv) survival after COVID-19 infection. Again, people with COPD did not display significant differences in PRS for any of the outcomes when

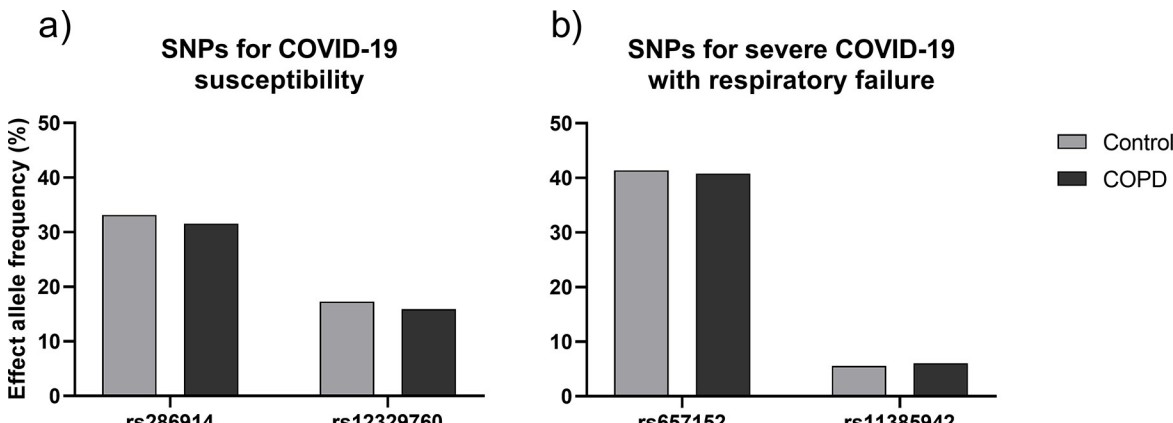

**Fig 1.** Allele frequencies for significant SNPs, for a) susceptibility (rs286914 and rs12329760) and b) severe response (rs657152 and rs11385942) to COVID-19 infection. No significant differences were found between COPD and control groups for any of the tested SNPs; rs286914: p-value = 0.60; rs12329760: p-value = 0.55; rs657152: p-value = 0.85; rs11385942: p-value = 0.72.

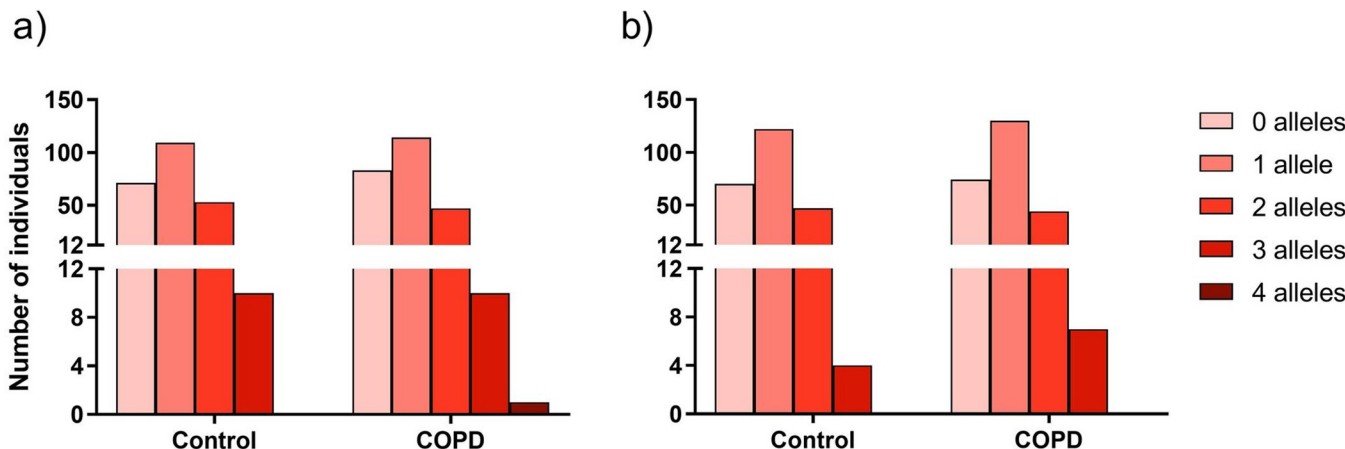

**Fig 2.** Number of people with a cumulative number of risk alleles for a) susceptibility for COVID-19 infection and b) severe COVID-19 with respiratory failure. 0 to 4 represents the sum of effect alleles for each COVID-19 associated phenotype. No significant differences were found, between COPD and control groups, for the distribution of people in the risk groups. a) p-value = 0.71 and b) p-value = 0.80.

compared to the control group, (Fig 3A–3D). These results were also validated by comparing both groups with the Minho cohort, with no significative differences (data not shown).

## Worldwide COVID-19 genetic risk scenario

**Single loci risk assessment.** We conducted a similar risk assessment, but focusing on the major human populations, as characterized in the National Center for Biotechnology Information (NCBI) [22,23], to determine how stable the COVID-19 associated genetic variants were in the human population. Overall, we found many significant differences across human populations. The effect allele frequency of SNP rs286914 in the European population (26.81%) was significantly lower than for the Portuguese (32.10%), African (38.01%) and American (35.90%) populations, but significantly higher than the Asian population (13.14%) (all p-values < 0.0001), (Fig 4A). The European effect allele frequency of SNP rs12329760 (27.23%) was significantly higher than in the Portuguese (19.90%) and American (17.2%) populations, but significantly lower than the African (28.84%) and the Asian (41.72%) populations (all p-values < 0.0001) (Fig 4B). There was no information available for SNPs rs286914 and rs12329760 in the Spanish and Italian populations, therefore, precluding the analyses. SNP rs657152 appears to have a decreased frequency in the European population (37.09%) when compared with the Portuguese (42.70%; p-value < 0.0001), Italian (38.43%; p-value = 0.0025), African (45.08%; p-value < 0.0001) and Asian (72%; p-value < 0.0001) populations. On the other hand, it has an increased frequency compared to the American (22.65%; p-value < 0.0001) population. No differences were found between European and Spanish (37.41%; p-value = 0.383) populations, (Fig 4C). For the last SNP of interest, rs11385942, the effect allele frequency on the European population (7.87%) was significantly higher than the Spanish (6%; p-value < 0.0001), African (5.76%; p-value < 0.0001), American (5.8%; p-value = 0.0018) and Asian (0.06%; p-value < 0.0001) populations, but significantly lower than the Italian population (10.33%; p-value < 0.0001). There were no differences between the European and Portuguese (5.56%; p-value = 0.06) populations, (Fig 4D). All data is detailed in S4 Table.

**Bi-allelic risk assessment.** A bi-allelic genetic analysis was performed to assess the number of people from each population that might have an aggravated genetic risk for COVID-19

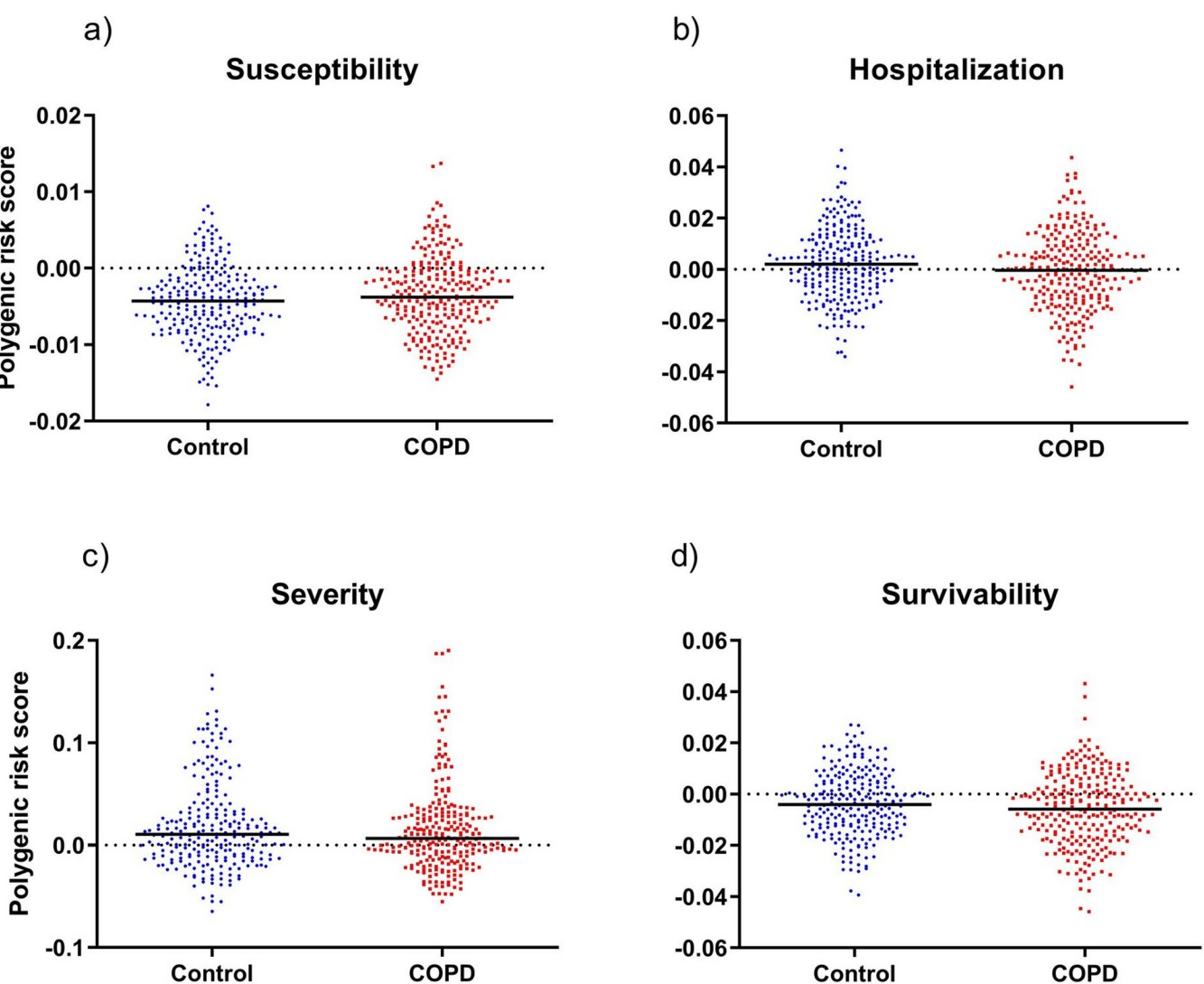

**Fig 3.** Polygenic risk score for susceptibility to COVID-19 infection (a), hospitalization due to COVID-19 infection (b), severe COVID-19 with respiratory failure (c) and survivability to COVID-19 infection (d). Results shown as scatter dot plots with median represented. No significant differences were found between the polygenic risk scores of COPD and control groups for any of the phenotypes tested; a) p-value = 0.05; b) p-value = 0.03; c) p-value = 0.38; d) p-value = 0.09.

infection or for severe COVID-19 with respiratory failure. Since this analysis multiplies the individual allelic frequencies that were already so diverse, the bi-allelic risk scenario is also highly heterogeneous among human populations. The European population was significantly different from all other populations in both COVID-19 susceptibility and severe COVID-19 with respiratory failure (p-value<0.0001 for all comparisons), (Fig 5A and 5B respectively). There was no information available for SNPs rs286914 and rs12329760 in the Spanish and Italian populations, therefore, precluding the estimation. The residues analysis highlights the African population as having the highest genetic risk for COVID-19 infection susceptibility (Fig 6A), with 12017 people per million estimated to be double-homozygous for the risk SNPs (total of 4 risk alleles), at the expense of the number of people with 0 risk alleles, which was only 194588 per million (S5 Table).

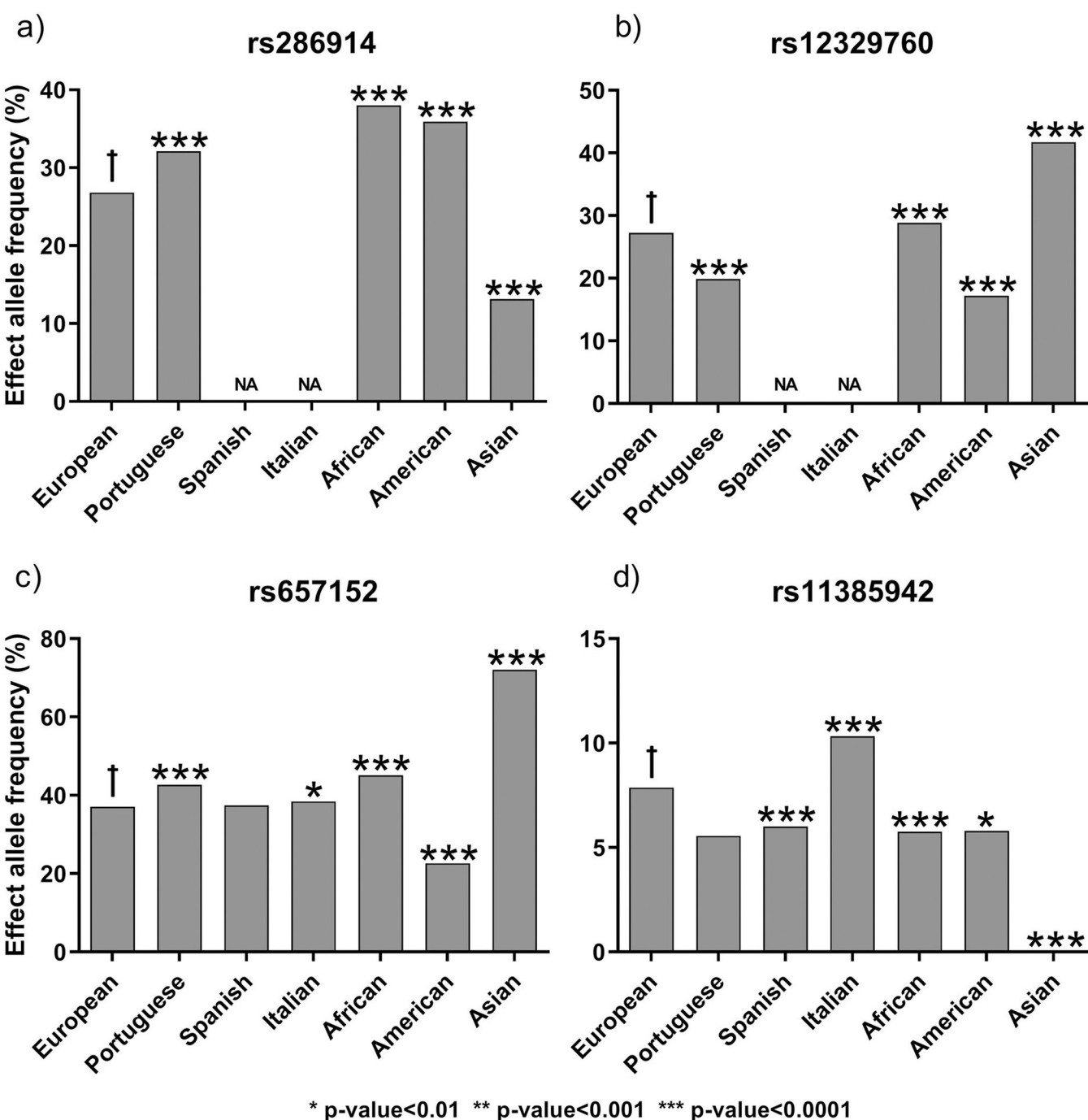

**Fig 4.** Effect allele proportions for COVID-19 susceptibility (a and b) and severe response to COVID-19 infection (c and d) SNPs in different world populations. Bars represent the proportion of each allele in the respective population. Allele frequencies for rs286914, rs12329760 and rs11385942 were obtained from gnomAD-Genome project, while rs657152 allele frequencies were obtained from the ALFA project. *: p-value<0.01; **: p-value<0.001; ***: p-value<0.0001;  - population used as reference for statistical analyses. NA—No data available.

The remaining populations displayed a higher resemblance in their risk distribution. As for the severe COVID-19 with respiratory failure, the Italian population had the highest estimated number of people, 1576 per million, being double-homozygous for the risk SNPs (total of 4 risk alleles) (Fig 6B). Alternatively, the American population had an enrichment of people

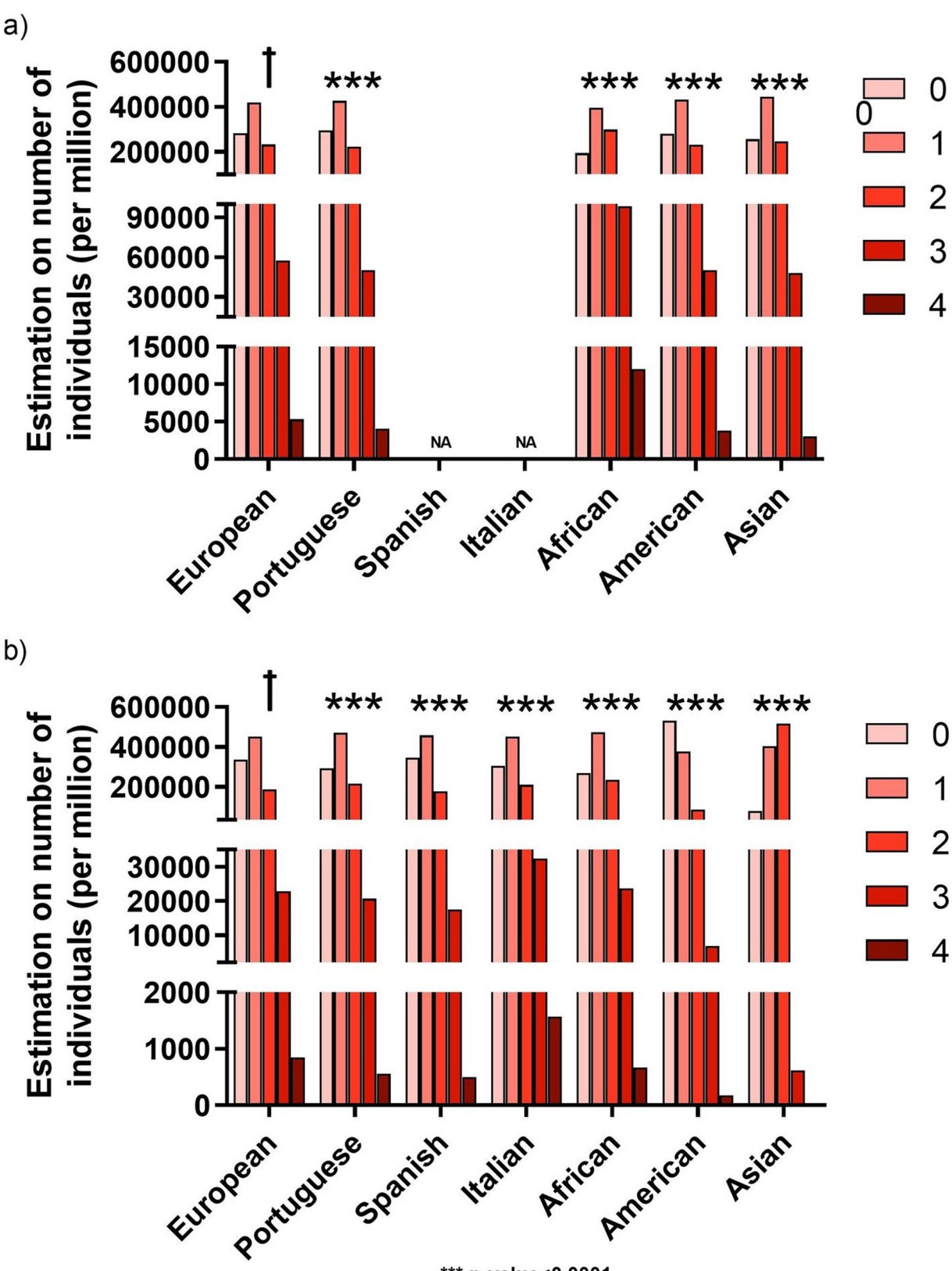

*** p-value<0.0001

**Fig 5.** Estimation on the number of people with a cumulative number of risk alleles in the world major populations for a) susceptibility for COVID-19 infection (rs286914 + rs12329760) and b) severe COVID-19 with respiratory failure (rs657152 + rs11385942). 0 to 4 represents the sum of effect alleles for each COVID-19 associated phenotype. Data for the Portuguese(n = 623), Spanish (n = 9761) and Italian (6363) populations correspond to observed values extrapolated to 1 million, whereas data for major world populations correspond to estimations (also to 1 million) based on the published effect allele frequencies, after Hardy-Weinberg equilibrium validation. Allele frequencies for rs286914, rs12329760 and rs11385942 were obtained from gnomAD-Genome project, while rs657152 allele frequencies were obtained from the ALFA project. All populations' distributions were compared with the European distribution.  - population used as reference for statistical analyses. ***: p-value<0.0001. NA—No data available.

with 0 risk alleles, 530912 per million. In addition, the Asian population had a substantial increase in the number of people with a total of 2 risk alleles, 518262 per million, when compared to the remaining populations, (S6 Table).

## Discussion

This study demonstrated that, genetics seem not to contribute to the increased risk of poor outcomes due to COVID-19 in people with COPD. High heterogeneity for COVID-19 associated genetic variants across world populations was also demonstrated. Comparative analyses on genetic predisposition to COVID-19 outcomes has been previously explored [29–31]. This study brings novelty by being the first comprehensive study (single-loci, bi-allelic and polygenic risk assessment) on the genetic predisposition of people with COPD to COVID-19 phenotypes; and by assessing the overall genetical risk distribution for COVID-19 in a global scenario.

### People with COPD do not have increased genetic risk for COVID-19 associated phenotypes

Significant differences for COVID-19 relevant SNPs between people with COPD and the control group were not found. This was observed consistently by looking at individual effect alleles coming from previously published genetic studies [3–6], additive effect of risk variants and by polygenic risk scores, for each phenotype. Furthermore, people with COPD could not be differentiated from controls when considering the genetic basis for COVID-19 susceptibility, neither for associated hospitalization, respiratory failure or death. These findings, together with recent reports showing that people with COPD have a poorer prognosis [8], support the hypothesis that the aggravated risk for poor COVID-19 clinical outcomes is independent of individual genetic backgrounds and is probably due to the characteristic debilitated state of these people's respiratory system [32], their chronic inflammatory state [32,33] and their increased number of respiratory infections [34], such as *pneumonia* [35].

### The world population shows high heterogeneity in COVID-19 genetic variants

Portuguese, Spanish and Italian populations belong to what is regarded as the European population, and therefore we would expect them to behave similarly regarding the COVID-19 associated SNPs, however, our results show otherwise. Differences were further emphasised when comparing the European population with the remaining world major populations, as observed in the bi-allelic risk for severe COVID-19 with respiratory failure, specifically in the American and Asian populations. The Italian population, which was the European COVID-19 first wave epicentre, showed the highest number of people double-homozygous for severe COVID-19 SNPs, highlighting a possible explanation for the severe epidemiological scenario observed in Italy. Overall, these results are in line with what has been reported in other studies, with the

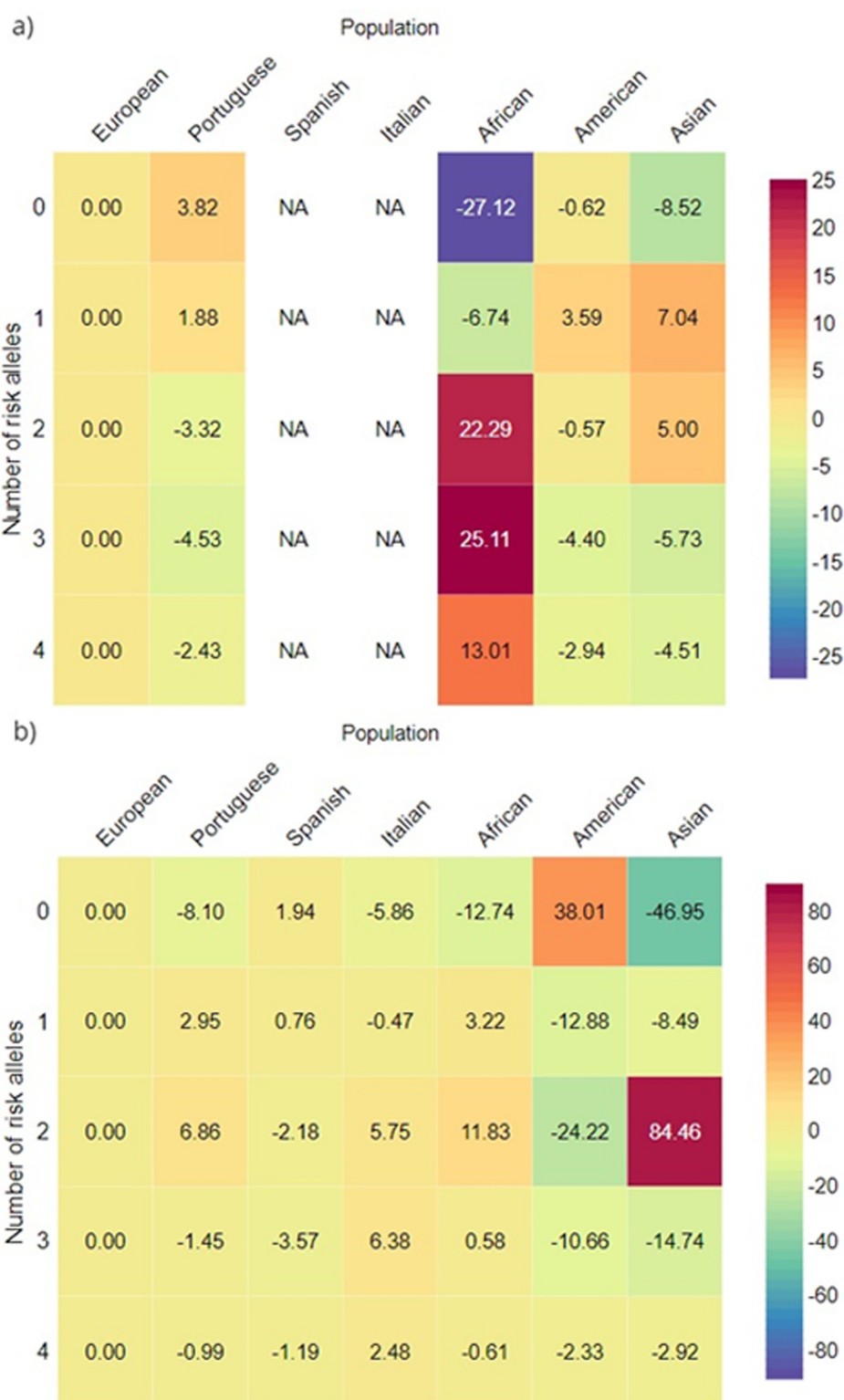

**Fig 6.** Adjusted residues heatmap for a) bi-allelic risk of COVID-19 infection susceptibility and b) bi-allelic risk of severe COVID-19 with respiratory failure. Red colour means that the target population/risk level is overrepresented, comparatively to the European distribution; Purple colour means that the target population/risk level is underrepresented, comparatively to the European distribution. NA—No data available.

risk for COVID-19 outcomes (cardiac, neurological, etc.) varying substantially between ethnicities [36].

## Study limitations

Our study was performed with a relatively small sample size (COPD: n = 255; control: n = 243; Minho: n = 380), although, in proportion to the whole Portuguese population, these numbers (0.006%) were close to those previously used for the Spanish and Italian populations study (0.021% and 0.011%, respectively) [3]. Rs41303171 may play a very important role in COVID-19 infection, knowing that this SNP is located in the gene encoding the most important SARS-CoV-2 receptor protein (ACE2) [37], however it could not be found in either of our cohorts, inhibiting its analysis. The COPD cohort was highly enriched on heavy smokers, whereas in the control group only a small proportion of people were smokers, which could have skewed the results. SNP rs11385942 was neither genotyped or imputed in our validation cohort (cohort from Minho), and therefore external validation of the results shown in Fig 2 was precluded. The PRS analysis was based on the summary statistics of external GWAS, meaning that any limitation of their studies would be translated into ours. SNP rs12329760 has only been associated with COVID-19 infection susceptibility indirectly, through the known role of its gene (*TMPRSS2*) in SARS-CoV-2 infection and not by a GWAS, therefore, lacking a genetic validation for such association. The populational risks (e.g. Fig 5) were estimated for a population of one million people, which led the Chi-Square test value to increase artificially. To downsize this limitation, we have used a very stringent significance threshold (p-value<0.01) and performed a residual analysis to highlight major effects within the dataset.

## Conclusion

This study showed that genetic background does not play a significant role in predisposition to severe COVID-19 among people with COPD based on the explored genetic variants. We also demonstrated the need to build a higher resolution European genetic map, so that differences in the distribution of relevant alleles, as those detected by us, can be easily accessed and used to better manage the diseases, ultimately, safeguarding populations with higher genetic predisposition to disease.

## Supporting information

**S1 Fig.** Allelic frequencies for significant SNPs, for A—susceptibility (rs286914 and rs12329760) and B—severe response (rs657152 and rs11385942) to COVID-19 infection. No significant differences were found between Control group of Baixo Vouga cohort and Minho cohort for any of the tested SNPs; rs286914: p-value = 0.54; rs12329760: p-value = 0.06; rs657152: p-value = 0.44. rs11385942 was not evaluated in Minho Cohort.
(PDF)

**S1 Table. COVID-19 associated SNPs description.** Ref. allele—reference allele. Alt. allele—alternative allele. A—adenine. C—cytosine. G—guanine. T—thymine. dupA—duplication of an adenine. V—valine. M—Methionine.
(PDF)

**S2 Table. Probability formula of having multiple risk alleles, assuming the Hardy-Weinberg's law.** SNP X and SNP Y are representative. p2—major allele in homozygosity probability; 2pq—heterozygosity probability; q2—minor allele in homozygosity probability.
(PDF)

**S3 Table. Sociodemographic, anthropometric and clinical characteristics of the Minho cohort.** N (%)—number of individuals and corresponding percentage; remaining data is presented as medians with interquartile range in square brackets. FEV1—Forced Expiratory Volume in 1-sec in litres; FVC—Forced Vital Capacity in litres; n.d.—no data available.
(PDF)

**S4 Table. Single-loci frequency comparison between European and the other world populations.** SNP—single nucleotide polymorphism; CHR:POS—genomic coordinates; N -total number of individuals enrolled in the study; A1—effect allele; A2—reference allele; A1F - Average effect allele frequency; A2F - Average reference allele frequency. Data source: rs286914, rs12329760 and rs11385942—gnomAD-Genome project7; rs657152—ALFA project6.
(PDF)

**S5 Table. Estimation on the number of people with a cumulative number of risk alleles in the world major populations, for susceptibility to COVID-19 infection (rs286914 + rs12329760).** 0 to 4 represent the sum of effect alleles. Data for the Portuguese population correspond to observed values (n = 623) extrapolated to 1 million, whereas data for major world populations correspond to estimations (also to 1 million) based on the published effect allele frequencies, after Hardy-Weinberg equilibrium validation. Allele frequencies were obtained from gnomAD-Genome project7.
(PDF)

**S6 Table. Estimation on the number of people with a cumulative number of risk alleles in the world major populations for severe COVID-19 with respiratory failure (rs657152 + rs11385942).** 0 to 4 represent the sum of effect alleles. Data for the Portuguese(n = 623), Spanish (n = 9761) and Italian (6363) populations correspond to observed values extrapolated to 1 million, whereas data for major world populations correspond to estimations (also to 1 million) based on the published effect allele frequencies, after Hardy-Weinberg equilibrium validation. Allele frequencies for rs11385942 were obtained from gnomAD-Genome project7, while rs657152 allele frequencies were obtained from the ALFA project6.
(PDF)

**S1 Raw data.**
(XLSX)

## Author Contributions

**Conceptualization:** Rui Marçalo, Alda Marques, Gabriela R. Moura.

**Data curation:** Rui Marçalo.

**Formal analysis:** Rui Marçalo.

**Funding acquisition:** Ana J. Rodrigues, Nuno Sousa, Manuel A. S. Santos, Alda Marques, Gabriela R. Moura.

**Investigation:** Alda Marques, Gabriela R. Moura.

**Methodology:** Rui Marçalo, Sonya Neto, Miguel Pinheiro, Alda Marques, Gabriela R. Moura.

**Project administration:** Nuno Sousa, Manuel A. S. Santos.

**Resources:** Nuno Sousa, Manuel A. S. Santos, Paula Simão, Carla Valente, Lília Andrade, Alda Marques, Gabriela R. Moura.

**Supervision:** Alda Marques, Gabriela R. Moura.

**Validation:** Alda Marques, Gabriela R. Moura.

**Visualization:** Rui Marçalo, Alda Marques, Gabriela R. Moura.

**Writing – original draft:** Rui Marçalo.

**Writing – review & editing:** Rui Marçalo, Sonya Neto, Miguel Pinheiro, Ana J. Rodrigues, Nuno Sousa, Manuel A. S. Santos, Paula Simão, Carla Valente, Lília Andrade, Alda Marques, Gabriela R. Moura.

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
