## [Decision Letter · Decision Letter 0]

2 Nov 2021

PONE-D-21-21399Genetic risk for COVID-19 outcomes in COPD and differences among worldwide populationsPLOS ONE

Dear Dr. Marçalo,

Thank you for submitting your manuscript to PLOS ONE. After careful consideration, we feel that it has merit but does not fully meet PLOS ONE’s publication criteria as it currently stands. Therefore, we invite you to submit a revised version of the manuscript that addresses the points raised by both reviewers during the review process. Specifically, both of them have major concerns on your work, as stated below.

We look forward to receiving your revised manuscript.

Kind regards,

Yan-Ming Xu

Academic Editor

PLOS ONE

Journal Requirements:

Reviewers' comments:

Reviewer's Responses to Questions

**Comments to the Author**

1. Is the manuscript technically sound, and do the data support the conclusions?

Reviewer #1: Yes

Reviewer #2: Partly

2. Has the statistical analysis been performed appropriately and rigorously? 

Reviewer #1: I Don't Know

Reviewer #2: N/A

3. Have the authors made all data underlying the findings in their manuscript fully available?

Reviewer #1: Yes

Reviewer #2: Yes

4. Is the manuscript presented in an intelligible fashion and written in standard English?

Reviewer #1: Yes

Reviewer #2: No

5. Review Comments to the Author

Reviewer #1: The current manuscript investigated the genetic risk (four selected SNPs) for COVID-19 in COPD patients, as well as in the worldwide populations. The manuscript is easy to read and has scientific merit. My concerns and suggestions for the authors are:

• Fig. S3 should not come before Fig. S1 and Fig. S2.

• The heatmap data of Fig. S1 and Fig. S2 can be considered as main figures.

• All figures can be improved by providing extra specific information. For example, in Fig. 1, title like “SNPs for COVID-19 susceptibility” or “SNPs for severe COVID-19 response” can be added; In Fig. 2, indicate what the “0, 1, 2, 3, 4” represent; In Fig. 4 and Fig. 5, a label such as “data not available” should be included on the Spanish and Italian groups instead of leaving them blank (as this may represent 0).

• Why write the p-values in the figure legends but not label them directly in the graphs?

• In Figure 5, how are the p-values being calculated (which data compared with which)?

• Make sure the references are in the correct format (e.g. ref. 10 and 11).

• rs41303171 was mentioned in the introduction as a genetic variant possibly playing a role in COVID-19 susceptibility. However, why is this SNP not tested? Especially when the gene of this SNP encodes the most important SARS-CoV-2 receptor protein, ACE2 (DOI: 10.1002/RMV.2122).

• It would be very helpful for the authors to include a table with SNP information (e.g. associated gene? Intron or exon variant? Missense or synonymous variant? Potential role in COVID-19 risk? etc.).

• The authors claimed that the “people with COPD do not have increased genetic risk for COVID-19”, however, it has to be made clear that this statement is based on the assessments of only the selected 4 SNPs and limited study samples.

• In addition to only concluding that “genetic backgrounds do not play a significant role in COVID-19 risk among people with COPD”, results obtained in Figures 4 and 5 are also very interesting, and should be further discussed/concluded. Indeed, several studies have shown that certain populations or ethnicities appear to have higher risk/severity of COVID-19 outcomes (e.g. DOI: 10.1017/cjn.2021.148).

Reviewer #2: The manuscript entitled "Genetic risk for COVID-19 outcomes in COPD and differences among worldwide populations" by Marcalo et al. I have the following concerns:

- Given that there is little genetic contribution for COVID-19 infection predisposition or worse outcomes observed in people with COPD, why are the authors not checking on populations with other lung-related diseases, such as asthma, ARDS, as well as tuberculosis?

- The title might be overstated and should be more specific on the ethnicity of the population being studied.

- The current title looks like as if there is genetic risk of COPD patients to COVID-19, this should be amended.

- For the SNPs already associated with susceptibility to COVID-19 infection/severe COVID-19 with respiratory failure, what exactly are these changes? E.g., any change of amino acids on encoded protein genes? These should be elaborated.

- It is unclear on the applications of the current work.

- Some of the figure details and text are too small to be seen.

- Typos and unfriendly mode of English usage can be found.

6. PLOS authors have the option to publish the peer review history of their article (what does this mean?). If published, this will include your full peer review and any attached files.

Reviewer #1: No

Reviewer #2: No

---

## [Author Response · Author response to Decision Letter 0]

18 Dec 2021

All comments from reviewers have been addressed and included in the revised manuscript.

---

## [Decision Letter · Decision Letter 1]

2 Feb 2022

Evaluation of the genetic risk for COVID-19 outcomes in COPD and differences among worldwide populations

PONE-D-21-21399R1

Dear Dr. Marçalo,

We’re pleased to inform you that your manuscript has been judged scientifically suitable for publication and will be formally accepted for publication once it meets all outstanding technical requirements.

Kind regards,

Yan-Ming Xu

Academic Editor

PLOS ONE

Additional Editor Comments (optional):

Reviewers' comments:

Reviewer's Responses to Questions

**Comments to the Author**

1. If the authors have adequately addressed your comments raised in a previous round of review and you feel that this manuscript is now acceptable for publication, you may indicate that here to bypass the “Comments to the Author” section, enter your conflict of interest statement in the “Confidential to Editor” section, and submit your "Accept" recommendation.

Reviewer #1: All comments have been addressed

Reviewer #2: All comments have been addressed

2. Is the manuscript technically sound, and do the data support the conclusions?

Reviewer #1: Yes

Reviewer #2: Partly

3. Has the statistical analysis been performed appropriately and rigorously? 

Reviewer #1: Yes

Reviewer #2: N/A

4. Have the authors made all data underlying the findings in their manuscript fully available?

Reviewer #1: Yes

Reviewer #2: Yes

5. Is the manuscript presented in an intelligible fashion and written in standard English?

Reviewer #1: Yes

Reviewer #2: Yes

6. Review Comments to the Author

Reviewer #1: The authors have satisfactorily addressed all my comments – I do not have further questions or concerns.

Reviewer #2: The authors have responded most of my comments. The manuscript is deemed to be acceptable for publication.

7. PLOS authors have the option to publish the peer review history of their article (what does this mean?). If published, this will include your full peer review and any attached files.

Reviewer #1: No

Reviewer #2: No

---

## [Editor Report · Acceptance letter]

11 Feb 2022

PONE-D-21-21399R1 

Evaluation of the genetic risk for COVID-19 outcomes in COPD and differences among worldwide populations 

Dear Dr. Marçalo:

I'm pleased to inform you that your manuscript has been deemed suitable for publication in PLOS ONE. Congratulations! Your manuscript is now with our production department. 

Kind regards, 

on behalf of

Dr. Yan-Ming Xu 

Academic Editor

PLOS ONE